# HUMAN-LEVEL CONTROL WITHOUT SERVER-GRADE HARDWARE

## ABSTRACT

Deep Q-Network (DQN) marked a major milestone for reinforcement learning, demonstrating for the first time that human-level control policies could be learned directly from raw visual inputs via reward maximization. Even years after its introduction, DQN remains highly relevant to the research community since many of its innovations have been adopted by successor methods. Nevertheless, despite significant hardware advances in the interim, DQN's original Atari 2600 experiments remain extremely costly to replicate in full. This poses an immense barrier to researchers who cannot afford state-of-the-art hardware or lack access to large-scale cloud computing resources. To facilitate improved access to deep reinforcement learning research, we introduce a DQN implementation that leverages a novel concurrent and synchronized execution framework designed to maximally utilize a heterogeneous CPU-GPU desktop system. With just one NVIDIA GeForce GTX 1080 GPU, our implementation reduces the training time of a 200-million-frame Atari experiment from 25 hours to just 9 hours. The ideas introduced in our paper should be generalizable to a large number of off-policy deep reinforcement learning methods.

## 1 INTRODUCTION

Reinforcement learning (Sutton & Barto, 2018) has long grappled with the ramifications of the Curse of Dimensionality (Bellman, 1966), a phenomenon in which exact solution methods become hopelessly intractable in the face of high-dimensional state spaces. As such, Deep Q-Network (DQN) (Mnih et al., 2013; 2015) was heralded as a landmark achievement for the field, establishing "deep" methods as a promising avenue for controlling environments that emit rich sensory observations. Through an effective combination of Q-Learning (Watkins, 1989) and a deep convolutional neural architecture (Krizhevsky et al., 2012), DQN became the first algorithm to achieve human-level performance on a majority of the Atari 2600 games when learning directly from raw pixel inputs. In contrast to previous efforts to integrate neural networks into reinforcement learning (e.g. Tesauro, 1992; Riedmiller, 2005), DQN proved to be robust, efficient, and scalable. Deep reinforcement learning has consequently become an active area of research in recent years.

Although DQN's performance on the Atari benchmark (Bellemare et al., 2013) has been surpassed by later methods (e.g. Hessel et al., 2018; Badia et al., 2020; Schrittwieser et al., 2020), the algorithm remains pertinent to ongoing deep reinforcement learning research. Its core elements—minibatched experience replay and a time-delayed target network—have been adopted by most off-policy deep reinforcement learning methods (e.g. Lillicrap et al., 2015; Fujimoto et al., 2018; Haarnoja et al., 2018). As such, the DQN algorithm is a good testbed for new ideas, as improvements to it are often directly transferable to state-of-the-art methods. Furthermore, its relatively straightforward implementation compared to modern successors, as well as its widely replicated results, have made it a reliable baseline for benchmarking and validating new methods. These factors have made DQN crucial to the research community even years after its introduction.

In spite of substantial improvements to computing hardware over nearly a decade, DQN is still expensive to train. The large computational cost stems from the need to conduct gradient-based optimization on a multimillion-parameter convolutional neural network. The original Atari 2600 experiments from Mnih et al. (2015), in particular, remain prohibitive for many to replicate. Agent training was conducted for 200 million frames on each of the 49 games considered, for a total of 9.8 billion frames (equivalent to about 5.2 years of real experience). Conducting the entirety of these experiments is utterly infeasible without access to costly Graphics Processing Units (GPUs), and can still take many weeks without access to a great number of them. This poses a significant barrier to deep reinforcement learning researchers, particularly those who lack substantial cloud computing resources. Given DQN's importance as a testbed and a baseline, this barrier puts a majority of researchers at an unfair disadvantage when it comes to publishing their ideas.

To foster improved accessibility to deep reinforcement learning research, we analyze the algorithmic structure of DQN and look for opportunities to reduce its runtime when executed on a standard CPU-GPU desktop system. In contrast to a separate line of inquiry into distributed DQN methods that focus on scaling to a large number of nodes (e.g. Nair et al., 2015; Ong et al., 2015; Horgan et al., 2018), we specifically consider the challenges of optimizing performance for a resource-constrained local system. We develop a modified DQN implementation based on a novel framework of Concurrent Training and Synchronized Execution; the framework is general and fits nicely into other target network-based methods as well. When trained on a single NVIDIA GeForce GTX 1080 GPU, our implementation reduces the runtime of a full Atari experiment (200 million frames) from 25 hours to just 9 hours compared to a highly optimized baseline DQN implementation. At this rate, all 49 experiments from Mnih et al. (2015) can be replicated in a relatively short timeframe using highly affordable, off-the-shelf hardware. Our implementation achieves human- and DQN-level performance on a large majority of the games. We plan to publicly release the code after the submission period to aid other researchers in reproducing these results quickly with limited hardware requirements.

## 2 BACKGROUND

DQN can be understood as the combination of Q-Learning and deep convolutional neural networks, along with supporting infrastructure to make this combination stable. The neural networks' generalization ability helps DQN learn effective control policies in the face of high-dimensional sensory inputs (e.g. images) where classic reinforcement learning and dynamic programming would be intractable. We provide a brief summary of the DQN algorithm in our notation (Section 2.1) and then establish assumptions about the CPU-GPU hardware model for which we will optimize its runtime (Section 2.2).

### 2.1 DQN AND NOTATION

In essence, DQN inherits the same fundamental objective of Q-Learning: to estimate a function $Q \colon \mathcal{S} \times \mathcal{A} \mapsto \mathbb{R}$ such that acting according to a greedy policy $\pi(s) = \operatorname{argmax}_a Q(s, a)$ maximizes the agent's expected discounted return $\mathbb{E}_\pi[\sum_{t=1}^{\infty} \gamma^{t-1} r_t]$ for some $\gamma \in [0, 1]$ (Sutton & Barto, 2018). Here, the environment is defined as a Markov Decision Process (MDP) of the form $(\mathcal{S}, \mathcal{A}, T, R)$. The sets $\mathcal{S}$ and $\mathcal{A}$ contain the environment states and agent actions, respectively, that are permissible in the decision process. The agent executes an action $a_t$ given the state $s_t$ at the current timestep $t$ and receives a scalar reward $r_t \coloneqq R(s_t, a_t)$, triggering a stochastic transition to a new state $s_{t+1} \in \mathcal{S}$ with probability $T(s_t, a_t, s_{t+1})$.

Whereas Q-Learning implements the Q-function as a lookup table, DQN implements it as a deep neural network parameterized by a vector $\theta$. Learning then amounts to a first-order minimization with respect to $\theta$ of a squared-error loss:

$$L(\theta) \coloneqq \frac{1}{2} \left( r_t + \gamma \max_{a' \in \mathcal{A}} Q(s_{t+1}, a'; \theta^-) - Q(s_t, a_t; \theta) \right)^2 \tag{1}$$

Rather than conducting updates immediately upon collecting a new experience (as Q-Learning does), DQN buffers each experience $(s_t, a_t, r_t, s_{t+1})$ in a replay memory $\mathcal{D}$. Every $F$ timesteps, the agent conducts a

gradient update on a minibatch of replayed experiences (Lin, 1992) sampled randomly from $\mathcal{D}$. This helps to circumvent the strong temporal correlations between successive experiences while also efficiently reusing samples (Mnih et al., 2015). For additional stability, the maximization in (1) is conducted using a stationary "target" network parameterized by a separate vector $\theta^-$. DQN updates the target network every $C$ timesteps by copying the parameters from the main network: i.e. $\theta^- \leftarrow \theta$.

Following Mnih et al. (2015), we assume that each action $a_t$ is selected according to an $\epsilon$-greedy policy (Sutton & Barto, 2018). That is, the agent selects the greedy action $\mathrm{argmax}_a Q(s_t, a)$ with probability $\epsilon_t \in [0, 1]$ and an action randomly from $\mathcal{A}$ otherwise. The $\epsilon$-greedy strategy linearly interpolates between the uniform-random and greedy policies, helping to balance exploration with exploitation (Sutton & Barto, 2018). In practice, it is common to start with $\epsilon_t = 1$ early in training and gradually reduce its value over time; we discuss the exact $\epsilon$-schedules used in our experiments in Section 5.

## 2.2 HARDWARE MODEL

Optimizing the performance of any algorithm necessarily requires some assumptions about the type of computer system on which it is being executed. In the interest of generality, we defer a discussion of our particular hardware specifications until Section 5. For now, it will be more useful to outline the general capabilities of the systems under our consideration here.

We define our abstract machine as a heterogeneous system that consists of two components: a Central Processing Unit ("CPU") and a coprocessor optimized for massively parallel computation ("GPU").[1] We assume that the GPU is suitable only for neural network operations: i.e. Q-value prediction (inference) and training (backpropagation). All other faculties are to be handled by the CPU, including but not limited to sampling from the environment, managing the replay memory, and preprocessing input data for the neural network. We also assume that the CPU is capable of executing $W$ program threads simultaneously.

As a result, the system can process up to $W$ CPU tasks and one GPU task in parallel. In practice, the original DQN algorithm only executes one task at a time, which is inefficient. The goal in the following sections is to modify the algorithm such that the machine's capabilities are fully utilized.

## 3 CONCURRENT TRAINING

The DQN algorithm repeatedly alternates between executing $F$ actions in its environment and then conducting a single training update on a minibatch of replayed experiences. While this is effective for finely interleaving data generation with learning, it is not efficient for the heterogeneous CPU-GPU systems we consider here. This is because either the CPU or the GPU is left idle at any given point in the process.

Ideally, we would fill these idle intervals by ensuring that a CPU- and GPU-intensive tasks are executed in parallel at all times. Unfortunately, the original DQN algorithm cannot be refactored to permit this possibility. The dilemma is that execution and training are sequentially dependent on each other. To see this, suppose that the agent has just completed a gradient update at some timestep $t$ that produces the current network parameters $\theta$. The DQN agent would now interact with its environment for $F$ steps, computing the Q-values $Q(s_i, a_i; \theta)$ for $i \in \{t, \ldots, t + F - 1\}$ in the process. The next update, which produces a new set of parameters $\theta'$, is scheduled to occur at the beginning of timestep $t + F$. Note that we could not have conducted this update any earlier, since the Q-values for action selection depended on $\theta$ and not $\theta'$. On the other hand, we cannot proceed with the next sequence of Q-values $Q(s_i, a_i; \theta')$ for $i \in \{t + F, \ldots, t + 2F - 1\}$

---

[1] For ease of presentation, we refer to any such coprocessor as a Graphics Processing Unit (GPU). GPUs are presently the most common form of hardware acceleration for machine learning research; they were used by Mnih et al. (2015) and are used for our experiments too (Section 5). Of course, other matrix-optimized Application-Specific Integrated Circuits (ASICs) would be equally suitable; these may become more prevalent in the future. The optimizations we propose here are sufficiently general to apply to any of these possible implementations.

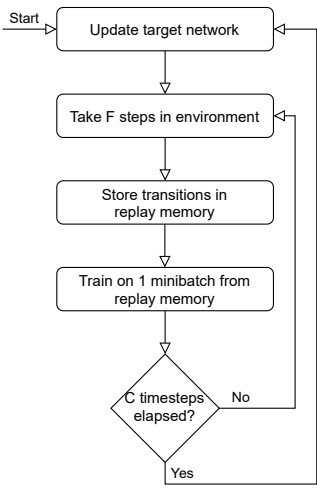

(a) Control flow for DQN.

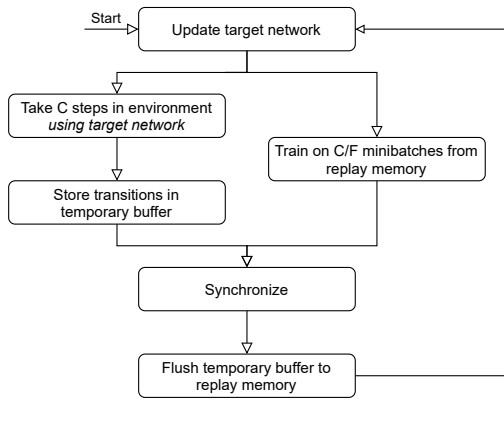

(b) Control flow for DQN with Concurrent Training.

Figure 1: Algorithm flowcharts for DQN (left) and DQN with our proposed Concurrent Training technique (right). By computing the agent's policy from the target network parameters $\theta^-$ instead of the main network parameters $\theta$, our method breaks the sequential dependency between training and environment sampling that is inherent to the standard DQN algorithm. This allows training and sampling to occur in parallel via multithreading, greatly reducing the overall runtime.

since these will depend on $\theta'$, which has not yet been prepared by the GPU. Sampling and training are interlocked processes as a consequence, making simultaneous execution impossible.

Rather than attempting to preserve DQN's original control flow, let us instead consider a slight modification to it. The sequential dependency in DQN arises from the fact that the training procedure needs to overwrite $\theta$ with $\theta'$, but the sampling procedure requires $\theta$ for action selection. If these tasks relied on different sets of parameters, then we would be able to parallelize these tasks. Quite fortunately, DQN already has a second parameter set available: its target network parameters $\theta^-$. We therefore propose to substitute $\theta^-$ as a surrogate for the main parameters $\theta$ during execution; that is, greedy actions are computed by $\arg\max_a Q(s, a; \theta^-)$ during $\epsilon$-greedy exploration. The mild assumption here is that a policy derived from $\theta^-$ should perform comparably to a policy derived from $\theta$. Given that $\theta^-$ is a time-delayed copy of $\theta$ that differs by at most $C$ timesteps of experience, this assumption should not be problematic in practice.

The result of this subtle change is that execution and training can now be decoupled (Figure 1), at least until the target parameters $\theta^-$ must be updated. Instead of sampling $F$ experiences and then training on one minibatch, it is possible to sample $C$ experiences while *concurrently* training on $C$ / $F$ minibatches using a separate program thread. Overall, the total computation is the same, but both the CPU and GPU are kept busy. After $C$ timesteps have elapsed, the two threads must synchronize to copy $\theta^-$ from $\theta$ as usual. To avoid a race condition between the threads, we also temporarily buffer the experiences collected by the sampler thread, transferring them to the replay memory $\mathcal{D}$ only when the threads are synchronized. This ensures that $\mathcal{D}$ does not change during training, which could produce non-deterministic results.

**Related Work**    We found no previous algorithms that utilize a similar technique to this type of concurrency, in which the target network parameters $\theta^-$ are used to break the dependency between sampling and training.

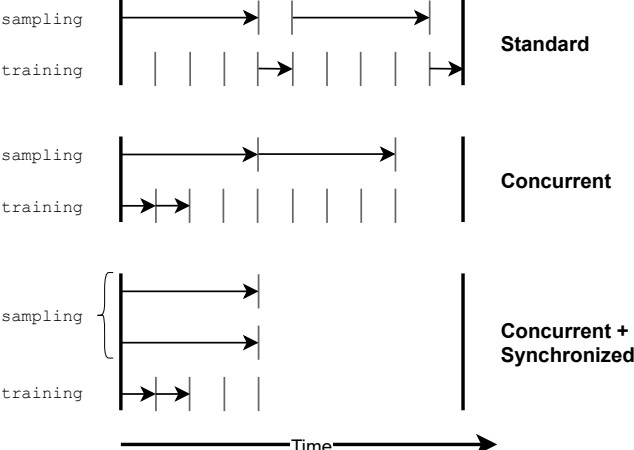

Figure 2: Abstract timing diagrams illustrating the theoretical speedup of our optimizations in DQN. By overlapping sampling and training, the training cost can be effectively masked (concurrency). To mitigate the remaining sampling bottleneck, multiple environments can be simulated in parallel (synchronization).

Daley & Amato (2019) first demonstrated that DQN with grouped minibatch training—where $C / F$ minibatches of training are conducted every $C$ timesteps—could learn effectively, where it was employed for the purposes of efficient $\lambda$-return calculation. However, unlike in our work, actions were still sampled using $\theta$ and training was not conducted concurrently. Distributed DQN methods that rely on a centralized training server (e.g. Nair et al., 2015; Ong et al., 2015; Horgan et al., 2018) could be viewed as an alternative form of concurrent training, since training and sampling occur simultaneously (although possibly on distinct physical systems). This type of concurrency is quite different from ours, since gradient computation is performed at the nodes, and the updates are applied to the parameters asynchronously and in a non-deterministic order. For these reasons, we do not expect that distributed methods would be efficient on a monolithic CPU-GPU system that we consider in our work (we elaborate on this point in Section 4).

## 4 SYNCHRONIZED EXECUTION

Concurrent training helps utilize the GPU more effectively by conducting training when the device would otherwise be idle. Assuming that sampling from the environment for $C$ timesteps takes longer than training on $C / F$ minibatches (our preliminary experiments indicated that this is the case), then concurrency effectively masks the computational cost of training. We illustrate this effect in Figure 2. Because training is not the performance bottleneck anymore, the algorithm's critical path has become the agent-environment interaction. To further reduce the overall runtime, we must turn our attention to techniques that allow the agent to gather samples more rapidly.

Since it is assumed that the environment's interaction speed cannot be optimized itself, a common approach for accelerating sample collection is to instantiate multiple (simulated) environments that can be updated in parallel on the CPU (e.g. Mnih et al., 2016). Based on our hardware assumptions in Section 2.2, up to $W$ environments can be updated at once. Simultaneously sampling from these instances reduces the average time spent collecting each individual sample (Figure 2). These so-called asynchronous methods do not rely on locks or other synchronization primitives when updating the parameters $\theta$, which can theoretically increase overall throughput. However, this creates other issues: updates are applied in a non-deterministic

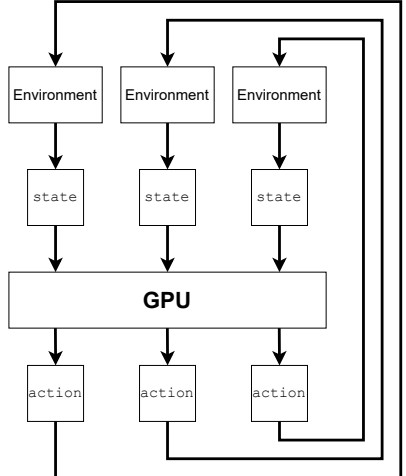 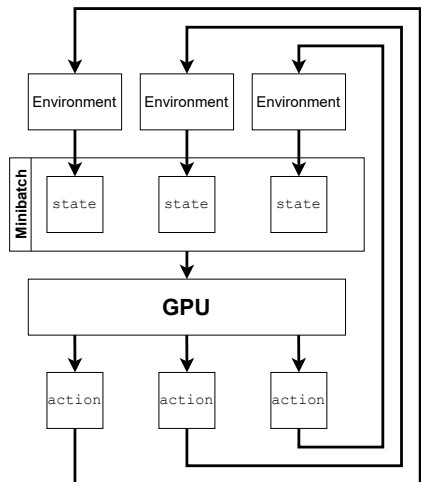

(a) Asynchronous execution.  Threads compete for limited device bandwidth, reducing throughput.

(b) Synchronized execution. Threads synchronize and share device transactions in a single minibatch.

Figure 3: Attempting to run asynchronous environments for deep reinforcement learning leads to significant resource competition between threads (left), introducing a performance bottleneck. This can be alleviated by first synchronizing the threads and then sharing computation in a single minibatch (right).

order, may be outdated with respect to the current parameters (due to latency), and may be clobbered by subsequent updates (thereby wasting samples).

When specifically targeting a monolithic CPU-GPU system, asynchronous learning encounters another problem regarding efficiency: the number of GPU transactions scales linearly with the number of samplers. Since the GPU has limited bandwidth, having too many threads attempting to communicate with it simultaneously can saturate the communication bus. Specifically, the samplers must compete for one shared GPU when computing their greedy actions for environment interaction, leading to diminishing improvements to runtime even when more parallel environments are added.

We propose to resolve this with a Synchronized Execution model (Figure 3). After the $W$ sampler threads each take a step in their respective environments, they are blocked by the main thread, which aggregates the environments' states into a single minibatch for efficient Q-value prediction on the GPU. The resulting Q-values are then distributed back to the sampler threads; the threads are subsequently unblocked. This process then repeats, with the sampler threads using the precomputed Q-values to select their actions in each iteration. In practice, the states and Q-values can be stored in shared memory arrays, meaning that no explicit message passing actually occurs between the threads. Synchronized Execution yields two major benefits. First, it exploits the massively parallel capabilities of the GPU by computing Q-values in a larger minibatch. Second, the total number of transactions between the CPU and GPU is reduced by a factor of $W$; in other words, this number is now independent of the number of samplers. While synchronizing the sampler threads every $W$ timesteps may incur a slight overhead cost, it is far outweighed by these computational benefits.

Synchronized Execution is orthogonal to Concurrent Training (Section 3), meaning that they can be combined into a single, fast DQN algorithm.  Altogether, the main program thread becomes responsible for orchestrating $W$ sampler threads and one trainer thread, for a total of $W + 1$ threads executing concurrently (the main thread does not perform computation). We outline pseudocode for this in Algorithm 1. For ease of

---

**Algorithm 1** DQN with Concurrent Training and Synchronized Execution (main thread)

---

Prepopulate replay memory $\mathcal{D}$ with $N$ experiences collected randomly
Initialize parameter vector $\theta$ randomly
Start $W$ sampler threads and one trainer thread
Create environment instance with state $s^{(j)}$ for each sampler thread, $j \in \{1, \ldots, W\}$
**for** $t = 1, 2, \ldots$ until convergence **do**
    **if** $t \bmod C = 1$ **then**
        Wait for trainer thread and all sampler threads to finish their tasks
        Flush all sampler threads' temporary buffers to $\mathcal{D}$
        Update target network parameters: $\theta^- \leftarrow \theta$
        Asynchronously dispatch trainer thread for $C \mathbin{/} F$ minibatches sampled from $\mathcal{D}$
    **end if**
    $i \leftarrow t \bmod W$
    **if** $i = 1$ **then**
        Wait for all sampler threads to finish their tasks
        In one minibatch, compute Q-values $Q(s^{(j)}, \cdot)$ for each sampler thread, $j \in \{1, \ldots, W\}$
    **end if**
    Asynchronously dispatch $i$-th sampler thread for one step in its environment using $Q(s^{(i)}, \cdot)$
**end for**

---

presentation, we omit the specific details of the sampler and trainer threads (which are identical to standard DQN but executed in parallel), focusing instead on the high-level dispatching logic.

**Related Work** The Synchronized Execution model in Section 4 is closely related to the strategy employed by PAAC (Clemente et al., 2017). PAAC is an online actor-critic method that similarly relies on shared memory arrays for efficient inter-thread communication during shared-minibatch synchronous execution. Stooke & Abbeel (2018) later applied this same form of synchronization to various deep reinforcement learning methods including DQN. In contrast to our work, training was conducted asynchronously via a distributed strategy across multiple GPUs, and the replay memory was statically partitioned across workers. We believe we are the first to integrate synchronized execution into a non-distributed, off-policy agent in which training minibatches are conducted in a fully deterministic order and retrieved from a globally shared replay memory. As an alternative to shared memory arrays, other parallel actor-critic methods like GA3C (Babaeizadeh et al., 2016) and Impala (Espeholt et al., 2018) have used queues to dynamically batch experiences, granting them enhanced robustness to slow or faltering workers, but also adding substantial complexity to the algorithms.

## 5 EXPERIMENTS

We design our experiments to answer two important questions regarding Algorithm 1. First, to what extent do Concurrent Training and Synchronized Execution reduce the runtime given fixed hardware specifications? Our hyperparameters of interest are whether these features are enabled, and how many sampling threads are created. Second, how does the fastest hyperparameter configuration's performance compare to the scores attained by DQN on the 49 Atari 2600 games tested by Mnih et al. (2015)?

To implement the Atari games, we use OpenAI Gym (Brockman et al., 2016) as an interface to the Arcade Learning Environment (ALE) (Bellemare et al., 2013). Our experiment setup is identical to that of Mnih et al. (2015), except where stated otherwise in Section 5.1.

### 5.1 SPEED TEST

We begin by conducting an ablation study on our fast DQN implementation, measuring the individual contribution of each component to the overall speed of the algorithm. We consider 14 algorithmic variants based

Table 1: Measured runtimes for DQN with/without concurrent training and/or synchronized execution as the number of sampling threads is varied. We report the mean and standard deviation over five trials (units: hours). All variants were evaluated on the same single-GPU system (see Section 5.1).

| Threads | Standard | Concurrent | Synchronized | Both |
|---------|----------|------------|--------------|------|
| 1 | $25.08 \pm 0.52$ | $20.64 \pm 0.29$ | — | — |
| 2 | $19.10 \pm 0.07$ | $14.00 \pm 0.00$ | $19.32 \pm 0.28$ | $14.72 \pm 0.13$ |
| 4 | $16.84 \pm 0.09$ | $12.14 \pm 0.05$ | $15.74 \pm 0.21$ | $11.08 \pm 0.08$ |
| 8 | $16.92 \pm 0.23$ | $11.68 \pm 0.04$ | $14.60 \pm 0.12$ | $9.02 \pm 0.16$ |

Table 2: Mean runtimes from Table 1 expressed as a percentage of DQN's mean runtime.

| Threads | Std. | Conc. | Sync. | Both |
|---------|------|-------|-------|------|
| 1 | 100.0% | 82.3% | — | — |
| 2 | 76.2% | 55.8% | 77.0% | 58.7% |
| 4 | 67.1% | 48.4% | 62.8% | 44.2% |
| 8 | 67.5% | 46.6% | 58.2% | 36.0% |

Table 3: Mean runtimes from Table 1 expressed as the speedup relative to DQN's mean runtime.

| Threads | Std. | Conc. | Sync. | Both |
|---------|------|-------|-------|------|
| 1 | $1.00\times$ | $1.22\times$ | — | — |
| 2 | $1.31\times$ | $1.79\times$ | $1.30\times$ | $1.70\times$ |
| 4 | $1.49\times$ | $2.07\times$ | $1.59\times$ | $2.26\times$ |
| 8 | $1.48\times$ | $2.15\times$ | $1.72\times$ | $2.78\times$ |

on whether Concurrent Training or Synchronized Execution is enabled and how many sampler threads are executed (chosen from $\{1, 2, 4, 8\}$). For now, we focus solely on wall-clock runtime for these experiments, deferring any performance evaluations until Section 5.2.

Our test hardware consists of a four-core (eight-thread) Intel Core i7-7700K CPU and an NVIDIA GeForce GTX 1080 GPU. Notably, this particular GPU model is not considered state of the art even for desktop hardware, having been released in 2016. As such, comparable or better hardware should be affordable to a large number of researchers and practitioners.

We choose Pong from the 49 Atari games as our test environment, although the choice of game is arbitrary and unlikely to impact the timing results. We make two slight changes to the experiment setup compared to that of Mnih et al. (2015). First, we train the agents for one million timesteps (instead of 50 million) and then multiply the results by a factor of 50. While this may increase the variance of the reported data, it makes the experiments much more feasible to implement. Second, we fix $\epsilon_t = 0.1$ for the entirety of this shorter training duration. This represents the greediest portion of the $\epsilon$-schedule used by Mnih et al. (2015) and, in turn, a worst-case scenario for GPU usage. Furthermore, this value of $\epsilon_t$ corresponds to the vast majority (98%) of the schedule, meaning that the results here closely reflect the full experiments.

We emphasize that our baseline DQN implementation has been heavily optimized in the interest of a fair comparison. All of our tested variations share the same core code—including time-critical portions like the deep neural network and replay memory. This means that any observable speedup is due only to Concurrent Training or Synchronized Execution. Our baseline appears to be significantly faster than popular existing DQN implementations (e.g. Roderick et al., 2017; Dhariwal et al., 2017; Hill et al., 2018) based on their reported results, although we did not compare them on our hardware. In any case, our proposed optimizations are equally applicable to all of these DQN agents.

For each hyperparameter combination, we report the mean runtime and standard deviation computed over five independent trials (Table 1). (Recall that these values were multiplied by 50.) We also alternatively express each mean as a percentage (Table 2) and as a speedup factor (Table 3) with respect to DQN's mean runtime, in order to help reveal trends in the data. We make several interesting observations. First, for any

number of sampler threads, enabling concurrency or synchronization is always helpful. Their benefits are synergistic too; enabling both is always faster than enabling just one. Interestingly, the reverse is not true; with both concurrency and synchronization disabled, increasing the number of sampler threads beyond four fails to further reduce the runtime. We hypothesize that this is due to intense GPU competition between the threads that we discussed in Section 4. Finally, and perhaps most importantly, the fastest combination is achieved by combining both features and maximizing the number of sampler threads, which almost triples the overall speed. These findings indicate that Concurrent Training and Synchronized Execution work together to enable full utilization of the system resources.

## 5.2 ATARI 2600 SUITE

We now evaluate the fastest hyperparameter configuration from Section 5.1 on all 49 Atari games, following the experimental procedures from Mnih et al. (2015) as closely as possible. The agents are trained for 50 million timesteps (200 million frames) in each game, and are periodically evaluated during training by executing an $\epsilon$-greedy policy ($\epsilon = 0.05$) for 30 episodes in a separate environment instance. We conduct the evaluation every 250,000 timesteps (Roderick et al., 2017) and report the best mean performance attained in each game (see Appendix A). Note that here we used four computers (each with multiple GPUs) to accelerate results collection, although each agent was still trained on exactly one GPU.

Out of the 49 games, our agent achieved human-level performance—i.e. at least 75% human-normalized score (Mnih et al., 2015)—in 33 games. Not only does this constitute a clear majority, but it also exceeds the 29 human-level scores achieved by DQN. Furthermore, our implementation outperformed DQN on 28 of the games—once again, a majority. We attribute the improved performance to the parallel environment instances, which help to diversify experiences via enhanced exploration while reducing correlations between experiences.

Overall, our implementation performed comparably to or better than DQN in 37 games. In the remaining 12 games where performance was not quite as good, our agent converged to relatively stable (but lower) scores roughly halfway through training. We speculate that this is due to the inadequate exploratory capabilities of the $\epsilon$-greedy policy. Indeed, several of these games like Private Eye and Venture are considered to be hard-exploration problems (Ostrovski et al., 2017); given that the scores for both DQN and our method consisted of only a single trial on each game, it is not surprising that the results would exhibit high variance due to random chance. Nevertheless, our strong performance across a large majority of the games gives us confidence that our implementation is effective for quickly learning human-level control policies on the Atari benchmark with modest hardware requirements.

## 6 CONCLUSION

We presented a fast implementation of DQN that is designed to efficiently utilize the resources of a CPU-GPU system. When executed on affordable desktop hardware, the implementation reduces the runtime of a full Atari experiment from 25 hours to just 9 hours—nearly triple the speed of our highly optimized DQN baseline. On top of this, the trained agent exceeded both human- and DQN-level performance in a majority of the 49 games tested.

We hope that the impact of our work will be twofold. First, once the code is released publicly, our implementation can function as a baseline or test environment for quickly validating new ideas in deep reinforcement learning. Second, our optimizations described here can inspire analogous adaptations to other important deep reinforcement learning baselines like DDPG (Lillicrap et al., 2015) and Rainbow (Hessel et al., 2018) for execution on desktop hardware. Continued efforts in this direction would further reduce computational barriers to research and be of great benefit to the community.

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

## A   ATARI 2600 RESULTS

Table 4 (next page) contains the best scores attained across all of the evaluation periods during training, on each of the 49 Atari games. The learners considered are Random, Human, and DQN (reproduced from Mnih et al. (2015)) as well as our fast DQN implementation. We also report the human-normalized performance for DQN and our method. The human-normalized ("norm.") scores are computed as $100 \times (\text{Score} - \text{Random}) / (\text{Human} - \text{Random})$. The higher human-normalized score between DQN and our implementation is emphasized (bold). Parenthetical values represent standard deviation calculated over the 30 evaluation episodes.

## B   HYPERPARAMETERS

Table 5 below summarizes the DQN-specific hyperparameters and their values used in all of our experiments. The networks were trained using "centered" RMSProp (Hinton et al., 2012) with a learning rate of $2.5 \times 10^{-4}$, first- and second-moment decay values of $0.95$, and a small constant of $0.01$ added to the denominator. These values are identical to those from Mnih et al. (2015), but are restated here for reference.

Table 5: DQN hyperparameters for all experiments.

| Hyperparameter | Symbol | Value |
|---|---|---|
| minibatch size | | 32 |
| replay memory capacity | | 1000000 |
| target update period | $C$ | 10000 |
| training period | $F$ | 4 |
| discount factor | $\gamma$ | 0.99 |
| replay memory prepopulation | $N$ | 50000 |

Table 4: Comparison of our fast DQN implementation with DQN on 49 Atari 2600 games.

| Game | Random | Human | DQN | Ours | DQN (norm.) | Ours (norm.) |
|---|---|---|---|---|---|---|
| Alien | 227.8 | 6875 | 3069 (1093) | 1319.0 (707.1) | **42.7%** | 16.4% |
| Amidar | 5.8 | 1676 | 739.5 (3024) | 1276.7 (210.4) | 43.9% | **76.1%** |
| Assault | 222.4 | 1496 | 3359 (775) | 3818.1 (866.8) | 246.3% | **282.3%** |
| Asterix | 210 | 8503 | 6012 (1744) | 11128.3 (6000.7) | 70.0% | **131.7%** |
| Asteroids | 719.1 | 13157 | 1629 (542) | 1475.3 (396.0) | **7.3%** | 6.1% |
| Atlantis | 12850 | 29028 | 85641 (17600) | 113926.7 (4267.8) | 449.9% | **624.8%** |
| Bank Heist | 14.2 | 734.4 | 429.7 (650) | 668.7 (82.0) | 57.7% | **90.9%** |
| Battle Zone | 2360 | 37800 | 26300 (7725) | 26300.0 (6309.0) | 67.6% | 67.6% |
| Beam Rider | 363.9 | 5775 | 6846 (1619) | 6538.9 (874.4) | **119.8%** | 114.1% |
| Bowling | 23.1 | 154.8 | 42.4 (88) | 57.1 (9.5) | 14.7% | **25.8%** |
| Boxing | 0.1 | 4.3 | 71.8 (8.4) | 96.1 (3.2) | 1707.1% | **2285.7%** |
| Breakout | 1.7 | 31.8 | 401.2 (26.9) | 373.4 (95.1) | **1327.2%** | 1234.9% |
| Centipede | 2091 | 11963 | 8309 (5237) | 5138.2 (2793.3) | **63.0%** | 30.9% |
| Chopper Command | 811 | 9882 | 6687 (2916) | 1973.3 (955.2) | **64.8%** | 12.8% |
| Crazy Climber | 10781 | 35411 | 114103 (22797) | 120300.0 (17662.2) | 419.5% | **444.7%** |
| Demon Attack | 152.1 | 3401 | 9711 (2406) | 10245.0 (1018.5) | 294.2% | **310.7%** |
| Double Dunk | -18.6 | -15.5 | -18.1 (2.6) | -15.6 (3.6) | 16.1% | **96.8%** |
| Enduro | 0 | 309.6 | 301.8 (24.6) | 343.6 (30.0) | 97.5% | **111.0%** |
| Fishing Derby | -91.7 | 5.5 | -0.8 (19.0) | 26.9 (10.2) | 93.5% | **122.0%** |
| Freeway | 0 | 29.6 | 30.3 (0.7) | 29.5 (1.1) | **102.4%** | 99.7% |
| Frostbite | 65.2 | 4335 | 328.3 (250.5) | 2606.7 (782.3) | 6.2% | **59.5%** |
| Gopher | 257.6 | 2321 | 8520 (3279) | 11566.7 (3261.3) | 400.4% | **548.1%** |
| Gravitar | 173 | 2672 | 306.7 (223.9) | 418.3 (312.0) | 5.4% | **9.8%** |
| H.E.R.O. | 1027 | 25763 | 19950 (158) | 22900.0 (2671.0) | 76.5% | **88.4%** |
| Ice Hockey | -11.2 | 0.9 | -1.6 (2.5) | -0.3 (1.8) | 79.3% | **90.1%** |
| James Bond | 29 | 406.7 | 576.7 (175.5) | 540.0 (124.8) | **145.0%** | 135.3% |
| Kangaroo | 52 | 3035 | 6740 (2959) | 9720.0 (1897.8) | 224.2% | **324.1%** |
| Krull | 1598 | 2395 | 3805 (1033) | 8327.5 (818.0) | 276.9% | **844.4%** |
| Kung-Fu Master | 258.5 | 22736 | 23270 (5955) | 33840.0 (5024.7) | 102.4% | **149.4%** |
| Montezuma's Revenge | 0 | 4367 | 0 (0) | 0.0 (0.0) | 0.0% | 0.0% |
| Ms. Pacman | 307.3 | 15693 | 2311 (525) | 3105.0 (1100.9) | 13.0% | **18.2%** |
| Name This Game | 2292 | 4076 | 7257 (547) | 7428.3 (716.1) | 278.3% | **287.9%** |
| Pong | -20.7 | 9.3 | 18.9 (1.3) | 18.7 (1.4) | **132.0%** | 131.3% |
| Private Eye | 24.9 | 69571 | 1788 (5473) | 103.3 (18.3) | **2.5%** | 0.1% |
| Q*Bert | 163.9 | 13455 | 10596 (3294) | 12236.7 (1979.1) | 78.5% | **90.8%** |
| River Raid | 1339 | 13513 | 8136 (1049) | 13405.0 (2655.5) | 57.3% | **99.1%** |
| Road Runner | 11.5 | 7845 | 18257 (4268) | 46873.3 (7020.8) | 232.9% | **598.2%** |
| Robotank | 2.2 | 11.9 | 51.6 (4.7) | 26.7 (3.7) | **509.3%** | 252.6% |
| Seaquest | 68.4 | 20182 | 5286 (1310) | 3397.3 (853.1) | **25.9%** | 16.6% |
| Space Invaders | 148 | 1652 | 1976 (893) | 1124.3 (247.1) | **121.5%** | 64.9% |
| Star Gunner | 664 | 10250 | 57997 (3152) | 12703.3 (4880.6) | **598.1%** | 125.6% |
| Tennis | -23.8 | -8.9 | -2.5 (1.9) | -1.0 (0.3) | 143.0% | **153.0%** |
| Time Pilot | 3568 | 5925 | 5947 (1600) | 6430.0 (1864.4) | 100.9% | **121.4%** |
| Tutankham | 11.4 | 167.6 | 186.7 (41.9) | 151.6 (60.8) | **112.2%** | 89.8% |
| Up and Down | 533.4 | 9082 | 8456 (3162) | 15653.7 (7689.6) | 92.7% | **176.9%** |
| Venture | 0 | 1188 | 380 (238.6) | 186.7 (135.8) | **32.0%** | 15.7% |
| Video Pinball | 16257 | 17298 | 42684 (16287) | 129182.8 (56706.7) | 2538.6% | **10847.8%** |
| Wizard of Wor | 563.5 | 4757 | 3393 (2019) | 1590.0 (522.8) | **67.5%** | 24.5% |
| Zaxxon | 32.5 | 9173 | 4977 (1235) | 3406.7 (652.3) | **54.1%** | 36.9% |

