# OpenReview forum: "Human-Level Control without Server-Grade Hardware"
_ICLR.cc/2022/Conference — ICLR 2022 Submitted_

### Official Review · Reviewer_Dq75 · 2021-10-18

**Correctness:** 3
**Technical Novelty And Significance:** 1
**Empirical Novelty And Significance:** 2
**Recommendation:** 3
**Confidence:** 4

**Main Review:**

# Main Review

The paper is well-written and well-motivated. The implementation appears to be much faster and uses reasonable methods to achieve this speedup. Having a performant DQN implementation that runs quickly and is publicly accessible would benefit the RL community. I think there are two main issues with the paper: the novelty and the empirical evaluation.

## Novelty

The first idea that the authors introduce, selecting actions from the target network to enable parallel acting and training, has not been used to speed up training to my knowledge. However, it is similar to the idea of Double Q-learning[1]. Additionally, multiple other researchers have introduced systems that allow for concurrent training and acting [2]. Thus, I would say that the novelty of this idea is limited.

The second idea, taking multiple actions in parallel, is a standard method as far as I know. I believe it e.g. is used in the stable baselines implementation. See e.g. (https://stable-baselines.readthedocs.io/en/master/guide/examples.html?highlight=vector#recurrent-policies) and (https://github.com/hill-a/stable-baselines/blob/master/stable_baselines/common/vec_env/subproc_vec_env.py#L51). To the best of my knowledge, the proposed method is not novel -- but please clarify how your proposals differ if there's a misunderstanding on my part.

[1] Deep Reinforcement Learning with Double Q-learning, 2016

[2] Sample Factory: Egocentric 3D Control from Pixels at 100000 FPS with Asynchronous Reinforcement Learning, 2020

## Empirical Evaluation

The authors seem to measure speedup of 25->9 hours on their own baseline DQN implementation. It does not seem reasonable to use a baseline you implement yourself when there are plenty of performant DQN implementations available online. Minor implementation details can often have outsized effect to outcome in RL, and the authors might not be incentivized to optimize the baseline as it's easier to improve upon a poor baseline. I would encourage the authors to start from a performant publicly available DQN implementation, e.g. the dopamine repo.

The authors write that “Our baseline appears to be significantly faster than popular existing DQN implementations (e.g. Roderick et al., 2017; Dhariwal et al., 2017; Hill et al., 2018) based on their reported results, although we did not compare them on our hardware.”. The difference might be entirely due to hardware differences, so this statement does not say much. Additionally, the cited papers are relatively old, and more modern implementations and improved hardware might be available today.

Table 4 contains no standard deviations. Are results just computed over 1 seed per environment? If so, it might not be statistically meaningful.

The code does not seem to have all dependencies listed (e.g. cv2 is missing), and there's also no example on how to use it. For a paper whose main contribution is on sharing a good implementation, this is not ideal.

## Minor comments

I would recommend that the authors show the learning curves (of e.g. averaged human-normalized scores) in the main paper.

I would encourage the authors to broadener their related work section to other methods that have been proposed for accelerating RL:

[3] Accelerating reinforcement learning through GPU atari emulation, 2019

[4] Large Batch Simulation for Deep Reinforcement Learning, 2021



**Summary Of The Paper:**

DQN is known to consume a lot of resources. This paper introduces an optimized version of DQN which speeds up training 25->9 hours. The authors use two main techniques to improve throughput. Firstly, the authors propose to select actions by computing the argmax over the target network. This decouples acting and training and allows these two operations to run asynchronously. Secondly, the authors compute actions for multiple environments in parallel on the GPU.

**Summary Of The Review:**

The paper aims for a reasonable goal and is both well written and well-motivated. However, the introduced methods are either well-known or incremental variations of known methods. Additionally, the empirical evaluation could be improved by using stronger baselines.

---

> ### Author Response · Authors · 2021-11-18
> **Clarifications regarding novelty and the empirical evaluation.**
>
> Thank you for the review. Addressing your concerns regarding novelty and the empirical evaluation,
>
> > “The first idea that the authors introduce, selecting actions from the target network to enable parallel acting and training, has not been used to speed up training to my knowledge. However, it is similar to the idea of Double Q-learning.”
>
> Double DQN uses the target network to select greedy actions from the main network for the purposes of bootstrapping. This is purely a training construct; the executed policy is still derived from the main network. Just because both methods use the target networks (but for different purposes) does not diminish the novelty of our approach. In fact, these strategies are orthogonal and could be easily combined.
>
>
> > “The second idea, taking multiple actions in parallel, is a standard method as far as I know. To the best of my knowledge, the proposed method is not novel -- but please clarify how your proposals differ if there's a misunderstanding on my part.”
>
> Indeed, parallel environments are not new, and our paper does not attempt to claim this as a contribution. In Section 4, we discussed how this is a common approach. Instead, the focus of our work is to compute actions for all of the environments simultaneously in a shared minibatch, thus reducing communication overhead between the CPU/GPU. The subtle difference here is the parallelization that occurs on the algorithm side, not the environment side. We do acknowledge the GA3C paper in our work, which applied a similar approach to an online actor-critic method (A3C), but to our knowledge this has not been done for an experience-replay method like DQN.
>
>
> > “Table 4 contains no standard deviations.”
>
> Table 4 does contain standard deviations; please see the parenthetical values next to the scores obtained by DQN and our implementation.
>
>
> > “The code does not seem to have all dependencies listed (e.g. cv2 is missing), and there's also no example on how to use it. For a paper whose main contribution is on sharing a good implementation, this is not ideal.”
>
> Our dependencies are `atari-py`, `gym`, `numpy`, `opencv-python`, and `tensorflow-gpu >= 2.0.0`. We apologize for the inconvenience; we removed our README (which contains usage information) for the sake of anonymity, and the requirements.txt file was not up to date.
>
>
> > “I would recommend that the authors show the learning curves (of e.g. averaged human-normalized scores) in the main paper.”
>
> This is a great idea; thank you for this suggestion. We can add this to the paper.

---

> > ### Comment · Reviewer_Dq75 · 2021-11-19
> > **Thanks for clarifications**
> >
> > Thanks for your clarifications! See below for an additional comment:
> >
> > *The subtle difference here is the parallelization that occurs on the algorithm side, not the environment side.*
> >
> > I'm aware of this distinction. My impression is that this type of parallelism is used in e.g. sample factory and stable baselines. Is that incorrect? If so I would be happy to update my review.

---

> > > ### Author Response · Authors · 2021-11-19
> > > **stable-baselines3 DQN does not support VecEnv, maybe other frameworks do though.**
> > >
> > > We tested DQN from stable-baselines3 (https://github.com/DLR-RM/stable-baselines3) using a VecEnv and received this error:
> > > `ValueError: Error: the model does not support multiple envs; it requires a single vectorized environment.`
> > >
> > > This online discussion appears to corroborate that stable-baselines3 does not support vectorization with DQN:
> > > https://githubmemory.com/repo/PettingZoo-Team/SuperSuit/issues/77
> > >
> > > Of course, this is just one framework; maybe other libraries do support vector environments for DQN. In general, we see your concern regarding the functional similarity between our approach and VecEnv. Since our implementation is based on round-robin dispatched threads that run asynchronously *except* when explicitly synchronized, we did not originally make this comparison.

---

> > > > ### Comment · Reviewer_Dq75 · 2021-11-19
> > > > **VecEnv**
> > > >
> > > > The fact that DQN does not support VecEnv does not imply that no other algorithms in stable-baselines do. In light of the error message, I think it's reasonable to assume that some do. I believe that the important question is whether the ones that do use the type of parallelism that you propose.
> > > >
> > > > For Sample Factory, the code in https://github.com/alex-petrenko/sample-factory/blob/master/sample_factory/algorithms/appo/policy_worker.py#L87 seems to indicate that multiple environments (since it iterates over env_idx) are used for one forward pass.
> > > >
> > > > I don't know enough about these two code bases to make a claim one way or another. If neither of these codebases uses the type of parallelism you propose, I don't know of any that would. In this case, I would be happy to update my review and increase the score.

---

> > > > > ### Comment · Reviewer_Dq75 · 2021-11-29
> > > > > **comment**
> > > > >
> > > > > I would like to know if the authors could motivate their comparison to previous work. As in my original review, one worry is that the speedups you see compared to other reported results could be entirely due to hardware differences. Note that the cited papers  (Roderick et al., 2017; Dhariwal et al., 2017; Hill et al., 2018)  are relatively old. Would you be able to comment on this? Thanks!

---

> > > > > > ### Author Response · Authors · 2021-12-02
> > > > > > **Reply to VecEnv + comment**
> > > > > >
> > > > > > Based on our conversations with the other reviewers, it seems that deep RL methods do commonly use the type of parallelism we described previously. It is less clear whether it has been used for DQN before, but the point seems to be somewhat moot, since introducing it to DQN for the first time would offer marginal novelty anyway.
> > > > > >
> > > > > > ---
> > > > > >
> > > > > > Regarding the comparison to previous work: We were able to find runtime results for Roderick et al. (2017). In the paper, they state that
> > > > > >
> > > > > > > Each training process took about 3 days for our implementation.
> > > > > >
> > > > > > and then later add that
> > > > > >
> > > > > > >  The setup on which we tested these implementations is using two NVIDIA GTX 980 TI graphics cards along with an Intel i7 processor.
> > > > > >
> > > > > > Our hardware also uses an Intel i7 processor, and only one NVIDIA GTX 1080 graphics card. (It is unclear whether Roderick et al. used both graphics cards for training or just one.) Our graphics card model is slightly newer, but it was also available back in 2017 when their paper was written. Since our DQN implementation finishes in just over 1 day compared to 3 days for Roderick et al., we believe that the speedup is mostly due to software differences and not hardware.
> > > > > >
> > > > > > We did not find runtime data for the other two citations, since OpenAI Baselines and Stable Baselines are both software libraries that don't have papers associated with them (as far as we know).

---

> > > > > > > ### Comment · Reviewer_Dq75 · 2021-12-02
> > > > > > > **comment**
> > > > > > >
> > > > > > > Thanks for your response and clarifications!
> > > > > > >
> > > > > > > *deep RL methods do commonly use the type of parallelism we described previously*
> > > > > > >
> > > > > > > I am glad you were able to get to the bottom of this question. I would recommend clarifying this point in the next update of the paper.
> > > > > > >
> > > > > > > *Our hardware also uses an Intel i7 processor, and only one NVIDIA GTX 1080 graphics card*
> > > > > > >
> > > > > > > There are several more complications here, e.g. CUDA versions and memory. But this is a reasonable approximation of fair comparison. I would recommend adding this info to the next update of the paper.

---

### Official Review · Reviewer_287J · 2021-10-25

**Correctness:** 2
**Technical Novelty And Significance:** 2
**Empirical Novelty And Significance:** 2
**Recommendation:** 5
**Confidence:** 4

**Main Review:**

strength: Nearly three times acceleration of the proposed framework is very attracting that facilitates the complex Atrai games playing with DQN on desktop-level computing resources.

weakness: It is more important to see how the proposed framework can shed light on the implementations of other DRL or even DL models. This important discussion has been largely ignored  in the manuscript. The generalization of the proposed method is highly expected.

**Summary Of The Paper:**

This paper introduces a DQN implementation to leverage a novel concurrent and synchronized execution framework for better utilizing a heterogeneous CPU-GPU desktop system. With one NVIDIA GeForce GTX 1080 GPU, the implementation reduces the training time of a 200-million-frame Atari experiment from 25 hours to just 9 hours.

**Summary Of The Review:**

The paper proposes an acceleration framework for DRL based on synchronized execution of GPU. The performance of the framework is attracting. However, instead of the sole showcase, it is more important to see how the proposed framework can generalize to more DRL tasks.

---

> ### Author Response · Authors · 2021-11-18
> **Concurrent Training/Synchronized Execution are applicable to almost any bootstrapping, experience-replay deep RL method.**
>
> Thank you for the review.
> Regarding the generalization of our framework, you stated
>
> > "It is more important to see how the proposed framework can shed light on the implementations of other DRL models. This important discussion has been largely ignored in the manuscript."
>
> We would like to point out that we do provide a brief discussion of when our techniques are applicable in the final paragraph of our conclusion.
> We can certainly expand and emphasize this discussion if you feel that it is necessary for the paper.
>
> In essence, any experience-replay method that estimates values by bootstrapping from a target network could adopt our methodology.
> This is because concurrent training only requires the use of a target network, and synchronized execution is applicable whenever environment instances can be simulated in parallel.
> Some popular examples that we mentioned in our text include DQN-like methods such as Rainbow, and continuous-control methods like DDPG, TD3, and SAC.

---

> > ### Comment · Reviewer_287J · 2021-11-23
> > **About the adaptability of the proposed method**
> >
> > Thank you for your response.
> >
> > Indeed the authors stated in the conclusion that "Second, our optimizations described here can inspire analogous adaptations to other important deep reinforcement learning baselines like DDPG (Lillicrap et al., 2015) and Rainbow (Hessel et al., 2018) for execution on desktop hardware." However,  "analogous adaptations” should be more specific and well defined.
> >
> > The main concern from the reviewer is that it is unclear who will be benefit from a specifc optimization of the computational execution of DQN.  If the proposed method is specifically designed for one DRL, it is questionable why DQN was selected for improvement as there are many more advanced DRL methods. If the proposed method is general and can be easily adapted to other DRLs, please give a technically illustrative disccusion. The authors stated in the second paragraph of the introduction section that this study on DQN was easy to adapt to other DRLs. However, it is not that natural to see how (at least the performance is not so easy to predict by the reviewer). It should be more specific.
> >
> > In general, from the reviewer's viewpoint, the proposed method is efficient but may not be so impactive unless it can be adapted to other more advanced DRLs and maintain the simialr performance. Similar concerns were also proposed by the first reviewer in the comment titled with "Retraining agents is a critical part of RL research".

---

### Official Review · Reviewer_pigN · 2021-11-02

**Correctness:** 3
**Technical Novelty And Significance:** 2
**Empirical Novelty And Significance:** 2
**Recommendation:** 5
**Confidence:** 3

**Main Review:**

Pros:
+ solves an actual technical/engineering problem that I encountered
+ reasonably clearly written, easy to read
+ well executed

Cons:
- changes the algorithms to be benchmarked slightly so it is not really fully equivalent to normal testing (using target network and multithreaded sampler)
- the speed up is not that impressive, if 25 hours is prohibitive I imagine that 9 might also be
- the ideas implemented aren't really that innovative or creative, I don't want to be arrogant but these are first things to try and I think I saw similar approaches but in context of synchronous A3C on gpu
- (not necessarily) the usefulness to the community hinges on the implementation of the framework and no details are mentioned


Comments/questions:
- The part where you run first 1mil instead 50mil steps is quite ironic, wasn't it easier to let it run for a bit longer and run it as is?
-"The dillema is that exectuion and training are sequentially dependent on each other" - are they really? Samples are put into replay memory and are randomly sampled every step. Considering that F is usually considerably bigger than the memory size it should not matter if we experience/train in sequence or in parallel. There is much bigger time delay between target and and original network and you explicitly ignore it.
- Figure 1: "Synchronize" step could be more explicit I guess.

**Summary Of The Paper:**

A new method for benchmarking RL algorithms like DQN, focusing on Atari suite and experiments from seminal DQN paper. The focus is to achieve speedup, allowing researchers with a modest setup (pc with gpu) to run the tests in reasonable time. The method combines concurrent data collection in multiple threads (and multiple environment instances) and training and achieves a speedup of x2.7 compared to vanilla DQN. At the same time the method slightly changes the nature of the algorithm by different action sampling (multiple env instances which is reminiscent of Asynchronous Methods paper from 2016) and slightly different action selection (using target network). The results are comparable to ones in the original paper overall, but vary on per game basis probably due to mentioned changes, slight change in hyper parameters and high variance of a single run.

**Summary Of The Review:**

The method proposed is not super innovative or creative but it's something that is needed and is well executed. It seems to do what it strived to do (speed-up) and there are no issues with the paper.

---

> ### Author Response · Authors · 2021-11-19
> **Responses to your comments/questions.**
>
> Thank you for the feedback. Responding to your comments/questions,
>
> > The part where you run first 1mil instead 50mil steps is quite ironic, wasn't it easier to let it run for a bit longer and run it as is?
>
> Given that we tested 14 different algorithmic variants (some of which are single-threaded baselines) for 5 trials each in Table 1, the time required to run all of these for 50M timesteps would be quite large. We chose instead to run them for a shorter duration and extrapolate the results, simply to make the experiments more feasible. Since we used a constant, worst-case epsilon value for exploration, these times should be representative of total training time. We also reflect the increased uncertainty of the extrapolation in the standard deviations.
>
> > "The dilemma is that execution and training are sequentially dependent on each other" - are they really? Samples are put into replay memory and are randomly sampled every step. Considering that F is usually considerably bigger than the memory size it should not matter if we experience/train in sequence or in parallel. There is much bigger time delay between target and original network and you explicitly ignore it.
>
> We think that you might be describing a form of asynchronous DQN, where sampling and training are completely decoupled from each other. (Please correct us if this is not what you mean.) The sequential dependency that we discuss in our paper is that new samples cannot be collected using the latest (“correct”) network parameters, until after the current minibatch has been trained on. While you could theoretically ignore this synchronization step, as you are suggesting, the algorithm would become nondeterministic. This is why we instead chose to use the target network for action selection, and synchronize the threads/parameters infrequently (whenever the target network must be updated, every 10k timesteps in practice).
>
> > Figure 1: "Synchronize" step could be more explicit I guess.
>
> Do you have any suggestions? Since Figure 1 is a flowchart, we felt that having the two parallel processes join together in a single block was already explicit. If you disagree, then please let us know how it could be improved.

---

### Official Review · Reviewer_KSo9 · 2021-11-02

**Correctness:** 3
**Technical Novelty And Significance:** 2
**Empirical Novelty And Significance:** 2
**Recommendation:** 6
**Confidence:** 4

**Main Review:**

irst and foremost, let me start by saying that I really appreciate the direction that the paper is taking and the problem that it is trying to solve. RL is often not optimized for academia-like hardware requirements, and we need more research and engineering work to figure out how to utilize these kinds of resources better. Kudos to the authors for working on this.

Now, onto the some of the issues that prevent me from recommending the paper for acceptance:

## Novelty

I am a little confused on how we should place this work in the literature. The novelty of the approach seems the clever combination of two frequently used ideas: (1) "vectorizing" environment executions so that the env interface can take $A = {a_0, a_1, ..., a_n}$ actions for $E = {e_0, e_1, ..., e_n}$ environments; (2) employing the target network to run inference on CPU whilst training a new set of parameters on default parameters on GPU from experience replay.

Point (1) however is an implementation details that is pretty much used across the large majority of modern DRL codebases out in the wild (e.g. ACME, stable-baselines, TorchBeast, ...), as in the majority of the cases environments are comparably much cheaper than the rest of the loop to run in parallel. This doesn't seem to be very well reflected in the manuscript (upon first read I was under the impression that this was something else, because I assumed that it was going to be something *beyond* VecEnvs).

Point (2) technically could be consider novel (AFAIK) when employed to produce this accelerator utilization scheme, but it is also a concept that exists in many existing libraries as an implementation detail. As the current manuscript is lacking in a strong discussion regarding (for instance) how using the target network affects DQN training (say, wrt. different kinds of MDPS -- lava world vs a pendulum optimizer -- or how the target network update frequency might affect the effectiveness of this trick), it feels odd to claim this as a strong contribution of the manuscript.


## Clarity of methods

I think the paper could use with a solid paragraph of definitions regarding what the authors mean when they say "distributed". One can technically run a distributed agent with extremely good results in local CPU-GPU hardware (see e.g. TorchBeast); it is true that the underlying complexity is greater, but many off-the-shelf frameworks abstract a lot of this away (and frankly speaking completely nail down the RPC/orchestration aspect of this problem -- see e.g. ACME recently). Since the manuscript is lacking a discussion on the tradeoffs that one makes when going that direction -- rather than sticking to a simpler threading / data communication model -- the comparisons with algorithms such as Impala (or R2D2) feel generally handwave-y.

I also found the explanation of the synchronization model to be a little unclear and underwhelming. Considering that these modifications are supposedly simple, I think a little bit of pseudo-python could go a long way towards making the whole methodology section clearer (particularly if other systems are also mentioned in the paper).


## Experiments

The experiments were also a little underwhelming. Firstly, for the speed test to be truly representative of reasoanble in-the-wild workloads, it probably should include comparisons against environments that have different step frequencies, models of wildly different sizes, the addition of recurrence (which is very common now in most RL-trained models), etc.

Secondly, I could not understand the point of the extrapolation done in the Pong experiments. Pong is very cheap to run, especially compared to running the whole Atari suite, so it feels bizarre to stop training early and then *linearly* (!) extrapolate the results. I would really like to understand why this procedure was chosen.

Thirdly, I think these sorts of empirical validations (for this particular paper) cannot rely on baselines numbers provided by papers, but rather be based on existing popular (and already well optimized) implementations of DQN. DQN is a relatively old method, and a lot of different public codebases have optimized it greatly with various amounts of tricks. It still remains unclear whether the presented tricks would actually make a significant increase in training speed when used on many of these implementations (I think at least one comparison should be attempted).

--

Please do note that I'd be very willing to increase my score provided that we (including the rest of the reviewers) have a productive discussions on how the manuscript can be quickly improved.

**Summary Of The Paper:**

The authors propose an implementation of DQN that focuses on maximising the data / training throughput in a common CPU-GPU machine setting. They do so by (asynchronously) running inference on multiple environments at once (and synchronously blocking on the env batch) and using the DQN target network parameters for inference rather than the default -- latest -- parameters of the network. They show considerable improvements in speed while producing comparable results with previous DQN results on the Atari suite.


**Summary Of The Review:**

The direction of the manuscript is commendable, and the ideas seem simple and cleverly executed. However there remain doubts wrt. novelty and on the validity of the experimental settings. Furthermore, the manuscript could also be significantly improved in terms of clarity.

---

EDIT: bumped score to 6.

---

> ### Author Response · Authors · 2021-11-19
> **Regarding novelty and clarity.**
>
> Thank you for the review. Regarding Novelty Point (1), we have an ongoing discussion with Reviewer Dq75 that is relevant to your concerns.
>
> In Point (2), you stated that our concurrent training is “a concept that exists in many existing libraries as an implementation detail.” Are you referring to concurrency in general as the “implementation detail”? We believe that our approach of using the target network to sample actions in parallel is novel.
>
> > I could not understand the point of the extrapolation done in the Pong experiments. Pong is very cheap to run, especially compared to running the whole Atari suite, so it feels bizarre to stop training early and then linearly (!) extrapolate the results. I would really like to understand why this procedure was chosen.
>
> As we said to Review pigN: Given that we tested 14 different algorithmic variants (some of which are single-threaded baselines) for 5 trials each in Table 1, the time required to run all of these for 50M timesteps would be quite large. We chose instead to run them for a shorter duration and extrapolate the results, simply to make the experiments more feasible. Since we used a constant, worst-case epsilon value for exploration, these times should be representative of total training time. We also reflect the increased uncertainty of the extrapolation in the standard deviations.
>
> It seems reasonable to assume that Pong would be computationally similar to the other games. Furthermore, why would linear extrapolation be problematic in this case?
>
> > I think the paper could use with a solid paragraph of definitions regarding what the authors mean when they say "distributed". One can technically run a distributed agent with extremely good results in local CPU-GPU hardware.
>
> This is a good point, and we agree that we should clarify on this in the paper. By “distributed,” we really mean along the lines of “asynchronous,” where different workers communicate updates to shared centralized parameters (which could be on the same device, like you said). Some issues with these methods is that updates are non-deterministic, and updates can be averaged together or randomly overwritten (depending on the implementation). We wanted to avoid this in our work to keep the results more true to the original DQN algorithm.

---

> > ### Comment · Reviewer_KSo9 · 2021-11-29
> > **Response**
> >
> > Thank you for your rebuttal, and apologies for the delay in submitting my response.
> >
> > Firstly, I looked at the discussions with the rest of the reviewers, and I am not convinced the manuscripts provide enough clarity wrt. the novelty of the ideas.
> >
> > Jointly calculating actions via an minibatch inference call is common at learning time: Acme, Seed RL, TorchBeast are some examples of recent open source RL agents that do this. There are many reasons these frameworks choose not to do this to collect training data, but the technique in itself is not new in the DRL landscape (and it is extremely common in general DL settings).
> >
> > I can however see how this could be considered novel in a "strict" DQN setting, however I think the manuscript could be improved if it were more precise about where the technique is currently used in the literature, and in what ways (and -- again -- it is used because of well justified reasons, so these could be used as arguments for this paper!), which would provide more solid grounds for sensible comparisons.
> >
> > Lastly, I do agree that utilising the target network for data generation is at the very least extremely uncommon across DQN variants, and I actually do like the idea overall, in this scheme.
> >
> > My primary concern with it is that it largely does feel like a complete hack. In the experimental space described by the tuple (ALE, feed-forward policy), utilising the target network to sample actions might be fine, but (a) DQN is now rarely used with simple FF policies, and (b) the choice of F and C become suddenly not only important for stability, but also for MDP exploration.
> >
> > We don't know how this will affect performance outside of this tuple, and DQN is very much the point of entry for people using RL *outside* of standard benchmarks, so this becomes especially important. As such I would expect an experimental section focuses on understanding this across multiple benchmarks (or at least across environment groups with known different training dynamics).
> >
> >
> > Regarding the pong experiments, upon re-reading the section I realised I misunderstood the point of that experiment, so I'm happy with it as is.
> >
> > Regarding the distributed bit, I really do think the wording here really matters, and asynchronous seems a fairly good alternative, so I would strongly suggest the authors to switch to it.
> >
> > ---
> >
> > Now, all of this said, I think I was being a little too harsh to the manuscript during review, and so I am bumping up my score, but my doubts do remain, and I think the manuscript could definitely be improved along the lines mentioned above.

---

> > > ### Author Response · Authors · 2021-12-02
> > > **Response 2**
> > >
> > > Thank you for the follow-up discussion, and for increasing your score. You have offered a number of important points for us to consider in an updated manuscript, which we appreciate.
> > >
> > > Your point about sampling with the target network is interesting: that stability has been coupled with exploration. We do not necessarily agree that this alone makes the method a "hack," but the implications on exploration are an important consideration for the paper. For example, it may be the case that 10k timesteps is short enough that the method works well for the ALE, but large C values would eventually lead to inadequate exploration. We think that the other factors you mention (non-feedforward model or a different F value) would have less of an impact on exploration --- nevertheless, we agree that testing different training dynamics would be better than speculating. Thanks again!

---

### Official Review · Reviewer_4Ciy · 2021-11-02

**Correctness:** 2
**Technical Novelty And Significance:** 2
**Empirical Novelty And Significance:** 2
**Recommendation:** 3
**Confidence:** 4

**Main Review:**

This paper is written clearly, and presents the main contributions in a straightforward manner. The inclusion of the code for reproducibility is appreciated.

However, there are a number of concerns with this paper, which are listed below.
- The motivation for doing this study is not very convincing; why would researchers need to retrain using DQN (or the variant of DQN introduced here) in Atari? Wouldn't just making the trained models available serve the same task?
- More generally, it is not convincing that a more efficient re-implementation of DQN (with a change to improve parallelism) is a notable contribution. If this concept could somehow be generalized to other domains, then this would potentially be useful. As it is, there are other algorithms that can achieve much better results than DQN, so it is less meaningful to implement this change. Also, how general are these results, and do they hold on different hardware?
- What does "real experience" refer to in Section 1, paragraph 2? Does this refer to a human playing?
- Section 1, paragraph 2 refers to GPUs as "costly", but this method still uses a GPU.
- The claimed speedup in terms of wall-clock time is 2.78x. It is not clear how significant this is, especially compared to other DQN implementations.
- Section 2: since this paper is about a more resource-efficient implementation of DQN, it is reasonable to assume that readers are familiar with DQN, so this discussion could be much more compact.
- Section 5.1 refers to "should be affordable" - it would be useful to have some kind of relative pricing - for instance, what would be the relative cost required to get an analogous speedup just by improving hardware?
- Section 5.2 refers to "all 49 Atari games"; there are actually many more than this, although 57 are commonly studied. There were indeed 49 games in the original DQN paper.


**Summary Of The Paper:**

This paper proposes a reimplementation of DQN that improves training speed with lower hardware requirements, showing a decrease of running a single Atari experiment from 25 hours to 9 hours on a specific system.

**Summary Of The Review:**

The main contribution of this paper is not sufficiently well-motivated or general; the re-implementation of DQN with more parallelism does not pose enough independent interest to be practically useful for most researchers. Suggestions for improvement would be to (1) show that this method could actually extend to settings beyond DQN and improve other RL algorithms; (2) perform a more in-depth hardware study comparing the cost to speedup via more powerful hardware vs. software improvements.

---

> ### Author Response · Authors · 2021-11-18
> **Retraining agents is a critical part of RL research.**
>
> Thank you for the feedback. Responding to your bullet points in the order they appear,
>
> 1. A crucial aspect of research is reproducibility. Making the trained models available, as you suggest, would only allow researchers to examine the final performance of the agents trained in our exact experimental setup. Retraining is useful because it allows experiments to be conducted that study how the agents’ learning is affected by differing conditions: for example, changes to the optimizer, return estimation, target network, replay memory, data preprocessing, or the environment (including possibilities other than Atari games). Having this capability is a necessary requisite for experimentation and science. The purpose of our work is not to simply provide pre-trained controllers for playing Atari games. Instead, it is to provide a test bench for researchers to quickly experiment with new ideas in deep reinforcement learning – especially those researchers who do not have unlimited computing resources available to them.
>
> 1. We chose to showcase DQN because it is a well-known deep reinforcement learning method that shares many features with newer algorithms (meaning that our proposed methods will generalize) and is commonly used as a baseline in publications (where it often must be retrained due to environmental or other implementation differences). In particular, any off-policy, experience-replay deep RL method that bootstraps from a target network can readily adopt the techniques we discuss in our paper. Possibilities include DDPG, TD3, and SAC.
>
> 1. “Real experience” refers to the total time taken when the games are played (possibly, but not necessarily, by a human) at the standard 60 Hz frame rate of the Atari 2600 console.
>
> 1. This is a bit of a semantic point. When we refer to “costly GPUs,” we mean that having access to a multitude of GPUs is costly, not that each individual GPU itself is costly. In fact, the GeForce GTX 1080 that we benchmark in our paper is not expensive (it is a desktop gaming model). We are happy to adjust the wording to clarify this.
>
> 1. The speedup of 2.78x is relative to our heavily optimized DQN implementation in TensorFlow/NumPy (but with concurrent/synchronous execution disabled) tested on the same hardware. We expect that applying our optimizations to other implementations would yield comparable speedups. An increase of 2.78x is quite significant given that the hardware is identical, and should be a considerable boon to researchers.
>
> 1. Regarding Section 2, we agree that the discussion of DQN could be shortened.
>
> 1. Showing the relative pricing for different hardware systems is a great idea. Regrettably, we will not have the time (or the resources) to include something like this for the conference.
>
> 1. With the phrase “all 49 games,” we are referring to all of the 49 games tested in the DQN paper.

---

> > ### Comment · Reviewer_4Ciy · 2021-11-20
> > **Re: Retraining agents is a critical part of RL research**
> >
> > Thank you for your response. My main concern is that the paper does not currently support the claims in your points 1, 2, and 5 above, namely:
> > - The comment in my review about making the models available was specifically in response to the statement in the paper that "The original Atari 2600 experiments from Mnih et al. (2015), in particular, remain prohibitive for many to replicate....Given DQN’s importance as a testbed and a baseline, this barrier puts a majority of researchers at an unfair disadvantage when it comes to publishing their ideas." If the goal is just to replicate a fixed algorithm (DQN or others) as a baseline, then making pre-trained models available would be useful for this purpose as well. It is not clear how the framework introduced in this paper would support changes to "the optimizer, return estimation, target network, replay memory, data preprocessing, or the environment" without major modifications to the code. Also, it is not clear that the speedup would still be possible (or what its magnitude would be) if those changes were made. Similarly, it is not clear that performance would not degrade if these changes were to be made.
> > - As mentioned in my main review, it would strengthen the paper substantially to actually demonstrate that this approach could achieve a speedup while maintaining performance for other algorithms.
> > - In the response, you mentioned that "We expect that applying our optimizations to other implementations would yield comparable speedups." Is there evidence in support of this?

---

### Decision · Program_Chairs · 2022-01-20

**Decision:**

Reject

**Comment:**

This paper introduces a variant of DQN optimized for desktop environments to make large scale experiments more feasible for anyone.

This paper was close. The reviewers appreciated the effort and motivation, but in the end the reviewers all seemed to think that the paper was not ready. The main contribution is framed as making DQN training more feasible, but the reviewers expected the paper to show examples of what the workflow for another architecture would look like and ideally present results for domains beyond Atari. In addition, several reviewers thought the paper could be more precise about (1) ruling out speed differences due to hardware and low-level software, and (2) contextualizing the speedups reported---does 3x matter, what should we expect, etc.

This is certainly an interesting direction. The AC personally thinks that if the authors take some steps to address the points above this will be a great and potentially high impact paper.